# The Domestic Dog as a Laboratory Host for *Brugia malayi*

**DOI:** 10.3390/pathogens11101073

**Published:** 2022-09-21

**Authors:** Christopher C. Evans, Katelin E. Greenway, Elyssa J. Campbell, Michael T. Dzimianski, Abdelmoneim Mansour, John W. McCall, Andrew R. Moorhead

**Affiliations:** 1Department of Infectious Diseases, College of Veterinary Medicine, University of Georgia, Athens, GA 30602, USA; 2TRS Labs Inc., Athens, GA 30604, USA

**Keywords:** dog, *Brugia malayi*, lymphatic filariasis, microfilariae

## Abstract

Of the three nematodes responsible for lymphatic filariasis in humans, only *Brugia malayi* is actively maintained in research settings owing to its viability in small animal hosts, principal among which is the domestic cat. While the microfilaremic feline host is necessary for propagation of parasites on any significant scale, this system is plagued by a number of challenges not as pronounced in canine filarial models. For this reason, we investigated the capacity in which dogs may serve as competent laboratory hosts for *B. malayi*. We infected a total of 20 dogs by subcutaneous injection of 500 *B. malayi* third-stage larvae (L3) in either a single (*n* = 10) or repeated infection events (125 L3 per week for four weeks; *n* = 10). Within each group, half of the individuals were injected in the inguinal region and half in the dorsum of the hind paw. To track the course of microfilaremia in this host, blood samples were examined by microscopy biweekly for two years following infection. Additionally, to identify cellular responses with potential value as predictors of patency, we measured peripheral blood leukocyte counts for the first year of infection. A total of 10 of 20 dogs developed detectable microfilaremia. Peak microfilaria density varied but attained levels useful for parasite propagation (median = 1933 mL^−1^; range: 33–9950 mL^−1^). Nine of these dogs remained patent at 104 weeks. A two-way ANOVA revealed no significant differences between infection groups in lifetime microfilaria production (*p* = 0.42), nor did regression analysis reveal any likely predictive relationships to leukocyte values. The results of this study demonstrate the competence of the dog as a host for *B. malayi* and its potential to serve in the laboratory role currently provided by the cat, while also clarifying the potential for zoonosis in filariasis-endemic regions.

## 1. Introduction

Lymphatic filariasis continues to be a significant cause of human morbidity and disability throughout tropical regions of the world, with an estimated 36 million living with chronic pathology resulting from infection with these mosquito-borne parasites. The causative agents are the filarial nematodes *Wuchereria bancrofti*, *Brugia malayi*, and *B. timori* [1]. Adult parasites occupy the afferent lymphatic vessels, where they can survive for an estimated 5–10 years and release microfilariae into the bloodstream for uptake and transmission by the mosquito vector [2,3]. Elimination efforts primarily depend upon mass drug administration strategies, however, the compounds in use (diethylcarbamazine citrate, albendazole, and ivermectin) are incompletely effective against adult parasites and are of little to no benefit to those with advanced pathology [4,5,6]. Therefore, the success of treatment relies largely on the ability of such drugs to suppress microfilaremia and prevent continued transmission [7].

One of the primary agents of lymphatic filariasis in south and southeast Asia is *B. malayi*; while only responsible for approximately 10% of human infections worldwide [1], it is commonly used as a laboratory model for lymphatic filariasis in general due to its ease of maintenance in small animals, including the domestic cat and Mongolian jird [8,9,10,11]. While *W. bancrofti* is responsible for the majority of total infections, this species has not been described to mature outside of primate hosts and, as such, is not amenable to investigation methods that require large quantities of parasite material [12,13,14].

The domestic cat has been favored as a model for lymphatic filariasis owing to the relatively high incidence and intensity of microfilaremia attained in experimental infections, allowing the collection of blood and development of third-stage larvae in laboratory-bred mosquitoes [9,11]. The Filariasis Research Reagent Resource Center (FR3; University of Georgia, Athens, GA, USA), which documents the infection of cats with *B. malayi* for parasite production and life cycle maintenance, has recorded that over the 12 years prior to this writing, 73 out of 134 infections developed to patency (i.e., presented with detectable microfilaremia; Table 1) with a median peak microfilaria concentration of 1950 mL^−1^. In subcutaneous infection of the cat, larvae are recoverable within the lymphatic vessels and lymph nodes proximal to the site of inoculation as early as 3 days post-infection, with the molt to the fourth larval stage occurring at 9–10 days and the molt to the adult stage occurring at 35–40 days post-infection. Adults largely remain confined to the major lymphatic vessels and their draining lymph nodes, where they mate and release microfilariae that ultimately arrive in the peripheral bloodstream [15,16]. The prepatent period in the feline host ranges from 70 to 147 days [9,17]. Microfilariae in peripheral blood exhibit a nocturnally sub-periodic pattern of changes in concentration, in which parasites are observed throughout the day and peak density occurs during nighttime hours [18,19].

Despite its apparent utility, the feline model for *B. malayi* presents some significant practical challenges. Restraining cats for venipuncture puts technicians at risk of injury by scratching and, significantly, at risk of bacterial infection in the event of a bite. When cats are subjected to such procedures on a routine basis, their disposition becomes less cooperative, further increasing risk to the handler. Moreover, the quantities of blood necessary for parasite collection usually require drawing from the largest superficial veins (e.g., the external jugular), meaning that safe and effective venipuncture typically requires the administration of anesthetics. Such anesthetics (e.g., ketamine hydrochloride) are strictly regulated and carry special requirements for acquisition and storage that must be managed by the laboratory. Venipuncture of the dog, however, typically presents none of the aforementioned challenges. For this reason, *B. pahangi*, a filarial parasite to which dogs are highly permissive, was once commonly utilized as the laboratory model for filariasis until granting agencies began encouraging the use of a species known to infect humans (i.e., *B. malayi*) to promote research more readily translatable to human medicine. Consequently, cats have become a necessary and expedient host for large-scale filariasis research over the past several decades due to their known-susceptible status. Prior to the current study, a thorough investigation of canine susceptibility to *B. malayi* had yet to be performed, and early attempts yielded mixed results.

In preliminary experiments performed by the FR3, a total of 9 dogs were infected by subcutaneous injection of *B. malayi* third-stage larvae (L3), of which 5 developed detectable microfilaremia. Each animal received a total of 1000 L3 (500 per side, inguinal) in this initial study. The duration of patency was relatively brief, never exceeding 10 weeks, and no animal attained concentrations greater than 1550 mL^−1^ (median peak concentration = 375 mL^−1^; data not shown). For a potential host to be considered suitable for maintenance of filarial worms, its microfilaremia must be chronic in nature. While the observed levels of microfilariae in these canine infections were too low to be of practical use, it was demonstrated that dogs were at least somewhat permissive to infection and the model warranted further study. The establishment of filarial parasites is influenced heavily by host specific factors, including immunity, and we expected that by increasing samples sizes we could mitigate the effects of these idiosyncrasies among individual animals.

In humans, infection with filarial worms produces a spectrum of pathological outcomes with two apparent extremes; the majority cases are asymptomatic and microfilaria-positive, while chronic pathology is more commonly associated with amicrofilaremia [20,21,22,23]. It is suggested that the parasites exert an immunomodulatory effect on host cells, resulting in these signs of either immune evasion or a failure thereof, respectively [24,25]. Alternative explanations, however, emphasize infection status over disease status as a reflection of the efficacy of the host immune response [26,27,28]. Additionally, asymptomatic ‘latent’ infections exist, which are defined by the presence of adults and the absence of microfilaremia [23,29]. Considering the apparent role of the host immune response in the establishment and pathogenesis of filarial infection, certain host factors may be investigated as early indicators of eventual microfilaremia status. Eosinophils, for example, function in both the innate and adaptive immune responses and are well-described responders to parasite infection. Peripheral blood eosinophil counts are typically elevated in infections with those helminths that require a tissue migration phase, including filarial nematodes and, as such, may be useful in diagnostic screening [30,31].

The same host diversity seen in *B. malayi* that makes it appealing as a laboratory model also represents a zoonotic potential. In endemic regions, the prevalence of microfilaremia in cats can reach as high as 20% [32,33]. While several studies have reportedly identified *B. malayi* microfilariae in dogs [34,35,36,37,38,39], the inherent limitations of conventional diagnostic techniques, in addition to the morphological and genetic similarity to other *Brugia* spp., leave some remaining uncertainty regarding the definitive competency of canine hosts and their role as potential filariasis reservoirs.

In the present study, we investigated the capacity to which the domestic dog may serve as a competent host for *B. malayi*, particularly with respect to microfilaria production in a laboratory setting. We tracked microfilaremia in 20 dogs experimentally infected with *B. malayi* over the first two years of infection with the aim of assessing the utility of this animal as a suitable laboratory host. Furthermore, to identify cellular responses to infection with potential value as predictors of eventual patency, which itself may improve the effectiveness of this model, we measured peripheral blood leukocyte levels during the first year of infection and tested for correlations with lifetime microfilaria production.

## 2. Results

### 2.1. Microfilaremia

Circulating microfilariae were first detected 12 weeks post-infection (Figure 1). A total of 10 out of 20 dogs developed detectable microfilaremia, with a detection limit of 33 microfilariae/mL. One of these patent animals only exhibited microfilaremia at a single timepoint before returning to undetectable levels (in the single foot inoculation group), while all of the other patent animals maintained patency for the remainder of the study (a minimum of 80 weeks from the first onset of detectable microfilaremia). Peak microfilaria density varied by animal (median = 1933 mL^−1^; range: 33–9950 mL^−1^). The greatest peak microfilaria density was observed in an animal in the inguinal trickle infection group. Equal numbers of male and female dogs developed patency (5 out of 10, each), and no differences were observed between sexes regarding lifetime microfilaria production (*p* = 0.13). The full microfilaremia dataset is available in Appendix A.

Lifetime microfilaria production for each dog was estimated by calculating area under the curve of microfilaria concentration over the duration of the study, represented as weeks·microfilaria/mL (wmf/mL). This value varied by animal (median = 70,200 wmf/mL; range = 9961–364,423 wmf/mL). A two-way ANOVA revealed no significant differences between infection groups (i.e., based on site and schedule of inoculation) in lifetime microfilaria production (*p* = 0.42).

### 2.2. Leukocyte Values

The peripheral concentrations of leukocytes (including total leukocytes, monocytes, lymphocytes, neutrophils, eosinophils, and basophils) are presented with a filaria-naïve baseline and for the first 52 weeks post-infection (Figure 2). During this period, median monocyte and neutrophil levels were slightly above normal reference range values (0.74 × 10^3^ and 8.7 × 10^3^ cells/µL, respectively), with all other median values remaining within reference ranges. Increased eosinophil concentrations from baseline were observed in 18 of 20 animals at 2 weeks post-infection (mean = 0.64 × 10^3^ cells/µL), with a later decrease at 4 weeks post-infection (mean = −0.44 × 10^3^ cells/µL); the timing of this elevated concentration appears to overlap the molt of *B. malayi* to the fourth larval stage. No clear relationship was observed between the mean of any of these leukocyte values and lifetime microfilaria production (total leukocyte R^2^ = 0.22; monocyte R^2^ = 0.045; lymphocyte R^2^ < 0.001; neutrophil R^2^ = 0.25; eosinophil R^2^ = 0.028; and basophil R^2^ = 0.42). The full CBC dataset is available in Appendix A.

### 2.3. Clinical Signs

During the course of the study, 3 out of the 20 dogs infected with *B. malayi* demonstrated temporary swelling in the hind limbs consistent with lymphedema that often accompanies infection with this parasite. One of the dogs (a female in the single foot injection group that remained patent from 12 weeks post-infection to the end of the study, with a peak microfilaremia of 2250 mL^−1^) exhibited intermittent swelling in both hind feet from 24 to 48 weeks post-infection before this resolved. Another dog (a female in the trickle inguinal injection group that never developed patency) exhibited swelling in the left hind foot from 10 to 13 weeks post-infection before resolving. Another dog (a female in the single foot infection group that showed patency at only a 20 weeks post-infection with a microfilaremia of 33 mL^−1^) exhibited swelling in both hind feet from 7 to 9 weeks post-infection before resolving. All observed swelling was characterized as mild to moderate and non-painful. No other clinical signs potentially related to infection were observed.

## 3. Discussion

In the present study, we tracked the course of microfilaremia in 20 dogs experimentally infected with *B. malayi* via multiple methods, as well as their peripheral blood leukocyte concentrations. Ten dogs developed detectable microfilaremia, nine of which remained patent for the remainder of the two-year study. In research settings, animals with high microfilaria concentrations are beneficial due to the quantities of parasites they produce without exceeding the volumetric limits of safe blood collection. Furthermore, a long-term microfilaremia with a duration on the order of years is highly desirable for continued parasite recovery and minimizing the number of host animals utilized. Large numbers of microfilariae are necessary for both direct study and propagation of the life cycle. In the field, animals with greater and longer-lived microfilaremia pose a relatively greater risk for zoonotic transmission [40]. Therefore, a thorough assessment and characterization of the course of *B. malayi* patency in the dog benefits work in both settings.

Beyond assessing the susceptibility of dogs to patent *B. malayi* infection, we performed the inoculation under four different conditions according to two variables in an attempt to identify an L3 injection route and schedule that optimized the establishment of parasites and microfilaremia. We employed one of two infection sites in each animal, injecting larvae subcutaneously into either the inguinal region or the dorsum of the hind foot with the expectation that the route of infection may influence the localization of adults and, therefore, their establishment and longevity within the host. In experimental infections of various mammalian hosts with *B. malayi* and *B. pahangi*, hind foot injections typically yield recoverable adults in the draining lymphatics of the injected limb, including the popliteal and inguinal lymph nodes [16,41,42]. Inguinal injections remain less localized, however, with adults frequently recovered from the heart and lungs in addition to the inguinal lymphatics and testes [43,44]. The present study revealed no differences in microfilaria production between dogs injected by either route. For each infection site, we also employed two different schedules of the infection process by performing either a single injection of L3 or a trickle infection of four weekly inoculations, with each procedure totaling 500 larvae per animal. While single injections are standard for experimental infections with filarial parasites, we included the trickle protocol under the premise of emulating the timing of natural infections, in which multiple infection events are more likely to occur; this may take advantage of immunomodulation effected by the earlier establishment of larvae; alternately, multiple challenges may prevent establishment due to host immune priming [45,46]. The present study revealed no significant differences in microfilaria production between either infection protocol, however the two dogs attaining the highest microfilaria concentrations were both inoculated by trickle infections, a finding that may hinge more directly on characteristics of the individual animals than the route of infection, but may warrant further investigation, nonetheless.

Regardless of infection protocol, the susceptibility of the dog to infection with *B. malayi* lends credence to its potential as an alternative host to the cat in this model of filariasis, if not a replacement. Canine venipuncture is a much simpler and safer procedure than in the feline model, eliminating the need for sedation and reducing the risk of injury to technicians; these are desirable improvements to the conventional model, which requires regular blood collection for the maintenance of the life cycle through vector blood-feeding. The replacement of the feline host requires that the dog perform similarly with respect to infection rates and microfilaria production or, at a minimum, that any deficiencies of the canine model can be justified by the improvements to handling and venipuncture procedures. In the present study, detectable microfilaremia was observed in 50% of dogs (*n* = 20), which is comparable to the 54% patency rate recorded by the FR3 in cats (*n* = 134; Table 1). This is less than the 70% reported in a similar study of *B. malayi* in cats (*n* = 10) [47], however, only 50% of cats demonstrated microfilaremia persisting more than two weeks, compared to 41% of dogs in the present study. Considering the sample sizes available, the rate of patency in cats and dogs appear to be reasonably similar.

The onset of patency in dogs ranged from 12–22 weeks after infection, or the first inoculation event in trickle infections. This is very similar to the 10–21-week prepatent period reported in cats [9,17]. All dogs that demonstrated microfilaremia at more than one timepoint remained patent for the duration of the study (i.e., a minimum of 82 weeks), while the median duration of *B. malayi* patency reported in cats is 44 weeks (range: 1–204 weeks) [47].

More than demonstrating patency, the usefulness of *B. malayi* in a laboratory host depends on the quantity of microfilariae available for study and propagation. In the present study, we observed a median peak microfilaria concentration of 1933 mL^−1^ in patent dogs (range: 33–9950 mL^−1^), while a recent study of cats observed a median of peak concentration of 1450 mL^−1^ (range: 25–6525 mL^−1^) [47]. Owing to a relatively greater available blood volume, a dog will always be a more productive source of parasites than a cat with similar microfilaria density. Consequently, our findings demonstrate that the dog may serve as an equivalent, if not more ideal, source of *B. malayi* microfilariae for research purposes than the conventional feline model.

By examining the establishment of *B. malayi* in the dog after experimentally induced infection, we have also clearly demonstrated the susceptibility of this species to the parasite in a controlled setting. While *B. malayi* microfilariae have previously been reported in natural infections of dogs [34,35,36,37,38,39], these cases occurred in environments in which an accurate diagnosis requires careful differentiation from other endemic filarial species to which canids are known to be permissive (e.g., *B. pahangi* and *B. ceylonensis*). Our findings, therefore, clearly confirm the susceptibility of the domestic dog to *B. malayi*, while the presence of microfilaremia supports its potential as a zoonotic reservoir that may require new consideration in filariasis elimination programs.

Of those dogs experimentally infected with *B. malayi*, only about half develop patent infections. If idiosyncrasies in host cellular immunity were predictive of the ultimate establishment of the parasite, they could be used to screen likely susceptible individuals for use as laboratory hosts. Such an establishment is influenced by the complex interaction of host and parasite factors. In general, strong T helper 2 (Th2) responses protect the host, while Th1 and Th17 responses yield immune-mediated pathology [48,49]. Meanwhile, a regulated, antigen-specific Th2 response is most often associated with long-lived, asymptomatic parasite survival [48,50,51]; in such an outcome, regulatory T cells and basophils are associated with host tolerance to infection while interleukin 4 (IL-4)-producing eosinophils and basophils maintain Th2 polarization [52,53,54,55]. Several host cell types (including monocytes, macrophages, lymphocytes, and dendritic cells) are purported targets of filarial worms for immunomodulation, and the presence of parasites has been shown to alter their ability to recognize and respond to infection [50,56,57,58,59].

The analysis of six components of cellular immunity in *B. malayi*-infected dogs (total leukocytes, monocytes, lymphocytes, neutrophils, eosinophils, and basophils) revealed no correlations between peripheral blood immune cell concentration and microfilaria production. Some variations during the course of early infection were observed, most notably an increase in eosinophil levels that may coincide with the third molt of the parasite. Increased eosinophil counts are also observed in cats and ferrets infected with *B. malayi*, though these spikes appear to coincide with the fourth molt and onset of microfilaremia, respectively [42,47,60]. While a more nuanced predictor of patency may yet be identified, our findings do not currently support any benefit through the pre-screening of dogs for *B. malayi* infection based on peripheral blood leukocyte counts alone.

## 4. Materials and Methods

### 4.1. Study Design

This study protocol was approved by the Institutional Animal Care and Use Committee at the University of Georgia (Protocol A2019 04-010).

A total of 20 beagle dogs (10 male, 10 female) were included in this study. Animal weights ranged from 6.1 to 10.0 kg at the beginning of the study. EDTA-anticoagulated blood samples were collected from study animals between 9:00 h and 11:00 h by jugular venipuncture for microfilaria counts and complete blood counts (CBCs). Baseline samples were collected immediately prior to infection. Blood samples were collected every 2 weeks for the 104 weeks following initial infection.

Animals were assigned to infection groups by block randomization accounting for sex and weight, with each group receiving one of the four infection methods described in the following section.

### 4.2. Experimental Infections

*Brugia malayi* (FR3 strain) infective third-stage larvae were obtained by dissection of *Aedes aegypti* (black-eyed Liverpool strain) mosquitoes fed 15 days prior on microfilaremic cat blood. Larvae were collected in Hanks’ balanced salt solution (Millipore Sigma, St. Louis, MO, USA) and washed in RPMI-1640 (Lonza, Walkersville, MD, USA) according to standard procedures (www.filariasiscenter.org/protocols; accessed on 29 March 2022). Twenty dogs were infected by subcutaneous injection of a total of 500 *B. malayi* third-stage larvae; parasites were administered in either a single inoculation (*n* = 10) or repeated inoculations (125 L3 per week for four weeks; *n* = 10). Within each of these groups, half of the animals were injected in the inguinal region and half in the dorsum of the hind paw (Table 2).

### 4.3. Microfilaria Counts by Giemsa Staining

Thick blood smears were prepared with heparin-anticoagulated whole blood collected from infected dogs by pipetting a 20-µL volume onto a glass microscopy slide, adding a 50-µL volume of water, and spreading this volume into a rectangular area 30 mm × 15 mm in size. Thick blood smears were dried at room temperature for a minimum of 24 h and heat fixed at 50 °C for 1 h before staining with a 10% Giemsa (VWR Scientific, Radnor, PA, USA) solution in TAE buffer. The total number of microfilariae per slide was counted under a compound microscope (Nikon Eclipse E200, Nikon Corporation, Tokyo, Japan) at 100× magnification. The average of three technical replicates for each sample was calculated and reported as microfilariae per mL.

### 4.4. CBC Analyses

On the day of infection and for the first year post-infection, CBC analysis was performed on peripheral blood samples from all of the study animals. EDTA-anticoagulated whole blood samples were collected from infected dogs and submitted for analysis at the University of Georgia Clinical Pathology Laboratory (Athens, GA, USA) utilizing an ADVIA 120 hematology analyzer (Siemens Medical Solutions USA, Inc., Malvern, PA, USA).

### 4.5. Statistical Analyses

Lifetime microfilaria production was calculated as area under the curve of microfilaria concentration over time for individual animals. A two-way ANOVA was performed to assess differences in microfilaria production based on infection method. The relationships between mean leukocyte concentration and lifetime microfilaria production were estimated by linear regression modelling. A Student’s *t*-test with two tails was performed to assess differences in lifetime microfilaria production between sexes of dog. All analyses were performed using GraphPad Prism version 8.0.1 for Windows (GraphPad Software, La Jolla, CA, USA).

## 5. Conclusions

The results presented here demonstrate the susceptibility of the dog to experimental infection with *B. malayi* via multiple methods. The rate, duration, and intensity of microfilaremia observed in this two-year study support the use of the dog as an alternative laboratory model in the role currently served by the cat. At present, no peripheral blood cellular immune components appear to be predictive of circulating microfilaria counts. By demonstrating patency, our findings also help elucidate the potential for canine reservoirs of *B. malayi* and zoonosis in endemic regions.

## Figures and Tables

**Figure 1 pathogens-11-01073-f001:**
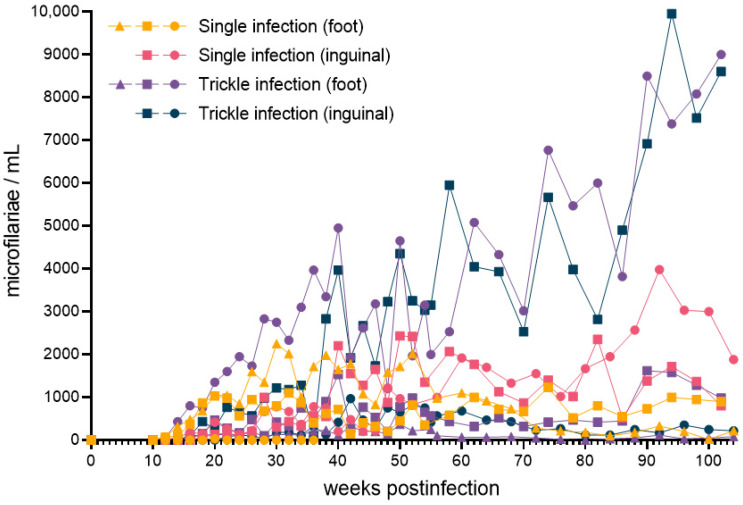
*Brugia malayi* microfilaria concentrations as determined by peripheral blood thick smear in experimentally infected dogs from initial infection to two years post-infection. Data for each dog developing detectable microfilaremia (*n* = 10) are presented. The route of subcutaneous infection was at either the dorsum of the foot or the inguinal region, comprising either a single bolus of L3 or a series of four weekly injections, each totaling 500 L3.

**Figure 2 pathogens-11-01073-f002:**
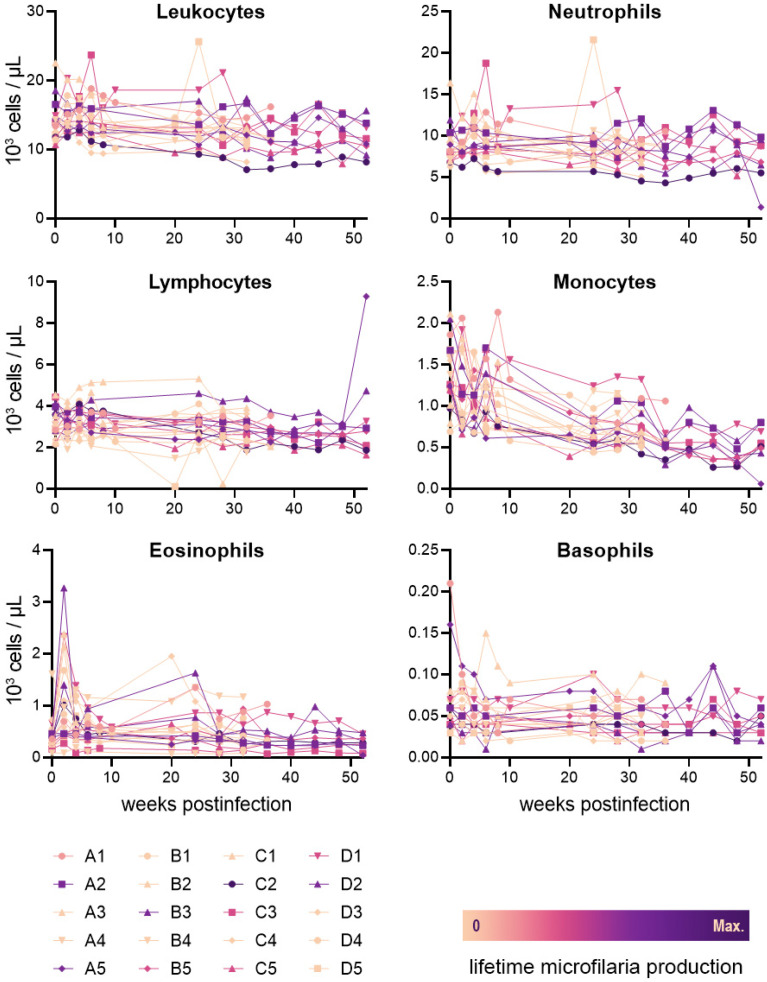
Peripheral blood leukocyte concentrations in dogs experimentally infected with *Brugia malayi*. The color of individual data sets reflects total lifetime microfilaria production for individual animals as calculated by area under the curve (range = 0–319,665 weeks·microfilaria/mL). Samples from the first timepoint were collected on the day of infection, prior infection.

**Table 1 pathogens-11-01073-t001:** Summary of experimental infections of cats with *Brugia malayi*. Data courtesy of the Filariasis Research Reagent Resource Center (FR3; University of Georgia, Athens, GA, USA).

Infection Group	Microfilaremia Status		Total
	+	−	
1	8	2	10
2	8	2	10
3	1	13	14
4	8	2	10
5	7	3	10
6	8	2	10
7	6	4	10
8	3	7	10
9	4	5	9
10	3	7	10
11	5	4	9
12	6	4	10
13	1	1	2
14	5	5	10
**Total**	**73**	**61**	**134**

Positive (+): microfilariae detected by peripheral blood thick smear; Negative (−): microfilariae not detected by peripheral blood thick smear.

**Table 2 pathogens-11-01073-t002:** Summary of methods for infecting dogs with *Brugia malayi*. Study animals received subcutaneous injections in either the inguinal region or dorsum of the hind paw, in either a single inoculation or a trickle infection spread over the course of four weeks. Five animals were infected by each method.

		Schedule of Infection	
		Single (1 injection of 500 L3)	Trickle (4 injections; 125 L3 each)
**Route of Infection**	Inguinal	*n* = 5	*n* = 5
	Hind Paw	*n* = 5	*n* = 5

## Data Availability

Data is contained within the article or Appendix A.

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
