# Peer review of "The Domestic Dog as a Laboratory Host for Brugia malayi"

_pathogens, 2022, doi:10.3390/pathogens11101073_

Round 1
Reviewer 1 Report
This is a well-written paper evaluating beagles as suitable hosts for Brugia malayi. Strengths of the study include 1) comparison between single and trickle infections, 2) long period (two years) of follow-up, and 3) frequent (every two weeks) quantification of circulating microfilaria counts. The paper shows that beagles develop patent infections about as frequently as cats (about 50% of the time), and that the degree of microfilaremia, as well as the duration, are both greater in dogs than has been historically observed in cats. Additionally, the authors found no differences in degree of microfilaremia between single and trickle infection groups (though the two animals that developed the greatest degree of microfilaremia were in the trickle group). While not providing new mechanistic scientific insights on the pathophysiology of filarial infections, this study is nonetheless an important contribution to the field of parasitology as it convincingly demonstrates that dogs can be suitable, and indeed possibly preferable, long-term hosts for Brugia for the purpose of having a supply of microfilariae for other studies.
Major critiques
1. Please include some information about the effects of Brugia infection on the health of the dogs. Did the dogs display any signs of illness (any lower extremity or scrotal swelling, fevers, behavioral changes, growth/weight changes, etc)?
Minor critiques
1. I may have missed it, but I do not recall a comparison of frequency of patency between male and female dogs. If not yet done, would conduct this analysis.
2. In the introduction (or in the discussion), would mention that having animals with chronic patent infection enables regular recovery of B. malayi mfs for the purposes of both A. direct study, and B. blood feeding to mosquitoes to enable generation of L3s for infection into other animal models. I think this background information would make the rationale for establishing long-term patent models more understandable for individuals that are not in the field of parasitology.
3. In the introduction, there is a statement that in the feline host larvae are found within the lymphatic vessel “proximal to the site of subcutaneous inoculation … “. Is this correct? My understanding was the L3 larvae after injection are typically.
4. In the introduction, the authors state that prelim experiments demonstrated relatively low mf concentrations in dogs after injection of Brugia malayi L3s. How many L3s were injected in those experiments? Given that the authors now find greater Mf numbers, any speculation as to why the prelim experiments found lower mf levels?
Author Response
We thank the reviewer for their very helpful feedback and critique. We have made revisions to the manuscript that we hope will address each of the issues raised and those are detailed below:
The reviewer raises an important issue in their major critique. We added a subsection in Results entitled "Clinical signs" to report our health observations on study animals. Briefly, three dogs developed temporary swelling in rear limbs during the course of the study, all of which resolved.
No differences were observed between sexes in either the number of dogs developing patency or their lifetime microfilaria production, and this was added to the results and methods sections.
We agree that the significance of long-term microfilaremia is an important point to make, and this was added to the introduction and discussion sections.
The third minor critique was cut off at the end, but I will try to address it. Larvae were not recovered after infection, so the timeline is informed by studies in reference 15 and 16 (Edeson and Buckley, 1959; Ewert, 1971). "Lymph nodes" was added to the sentence in addition to lymphatic vessels to be more inclusive of migration sites.
A total of 1,000 L3 were injected into each animal in the pilot experiment. We added this to the relevant part of the introduction section.
Reviewer 2 Report
This was a very well written and interesting article describing success in utilizing dogs as a host for experimental infection with brugia malayi. The authors also collected information regarding leukocyte levels post-infection and the relation to microfilaremia. I'd be interested to see similar information regarding more specific markers of inflammation, but understand that that is beyond the scope of this paper.
Author Response
We thank the reviewer for their feedback and critique. We agree that specific inflammatory markers may have revealed interesting and relevant findings related to infection establishment, but as they mention this was a procedure we could not adequately investigate within the focus of this study.